# Ticagrelor Increases *SIRT1* and *HES1* mRNA Levels in Peripheral Blood Cells from Patients with Stable Coronary Artery Disease and Chronic Obstructive Pulmonary Disease

**DOI:** 10.3390/ijms21051576

**Published:** 2020-02-25

**Authors:** Giorgio Aquila, Francesco Vieceli Dalla Sega, Luisa Marracino, Rita Pavasini, Laura Sofia Cardelli, Anna Piredda, Alessandra Scoccia, Valeria Martino, Francesca Fortini, Ilaria Bononi, Fernanda Martini, Marco Manfrini, Antonio Pannuti, Roberto Ferrari, Paola Rizzo, Gianluca Campo

**Affiliations:** 1Department of Medical Sciences, University of Ferrara, 44121 Ferrara, Italy; qlagrg@unife.it (G.A.); valeria.martino@student.unife.it (V.M.); 2Maria Cecilia Hospital, GVM Care & Research, 48033 Cotignola, Italy; vclfnc@unife.it (F.V.D.S.); frtfnc@unife.it (F.F.); mmanfrini@gvmnet.it (M.M.); fri@unife.it (R.F.); rzzpla@unife.it (P.R.); 3Department of Morphology, Surgery and Experimental Medicine and Laboratory for Technologies of Advanced Therapies (LTTA), University of Ferrara, 44121 Ferrara, Italy; luisa.marracino@student.unife.it (L.M.); ilaria.bononi@unife.it (I.B.); fernanda.martini@unife.it (F.M.); 4Cardiovascular Institute, Azienda Ospedaliero-Universitaria di Ferrara, 44124 Cona, Italy; pvsrti@unife.it (R.P.); laurasofia.cardelli@unife.it (L.S.C.); anna.piredda@unife.it (A.P.); alessandra.scoccia@unife.it (A.S.); 5University of Hawaii Cancer Center, University of Hawaii, Honolulu, HI 96813, USA; apannut@mac.com

**Keywords:** coronary artery disease (CAD), chronic obstructive pulmonary disease (COPD), Ticagrelor, Clopidogrel, Notch signaling, SIRT1

## Abstract

Ticagrelor is a powerful P2Y_12_ inhibitor with pleiotropic effects in the cardiovascular system. Consistently, we have reported that in patients with stable coronary artery disease (CAD) and concomitant chronic obstructive pulmonary disease (COPD) who underwent percutaneous coronary intervention (PCI), 1-month treatment with ticagrelor was superior in improving biological markers of endothelial function, compared with clopidogrel. The objective of this study was to investigate the mechanisms underlying these beneficial effects of ticagrelor by conducting molecular analyses of RNA isolated from peripheral blood cells of these patients. We determined mRNAs levels of markers of inflammation and oxidative stress, such as *RORγt* (T helper 17 cells marker), *FoxP3* (regulatory T cells marker), *NLRP3*, *ICAM1*, *SIRT1*, Notch ligands *JAG1* and *DLL4,* and *HES1*, a Notch target gene. We found that 1-month treatment with ticagrelor, but not clopidogrel, led to increased levels of *SIRT1* and *HES1* mRNAs. In patients treated with ticagrelor or clopidogrel, we observed a negative correlation among changes in both *SIRT1* and *HES1* mRNA and serum levels of Epidermal Growth Factor (EGF), a marker of endothelial dysfunction found to be reduced by ticagrelor treatment in our previous study. In conclusion, we report that in stable CAD/COPD patients ticagrelor positively regulates *HES1* and *SIRT1*, two genes playing a protective role in the context of inflammation and oxidative stress. Our observations confirm and expand previous studies showing that the beneficial effects of ticagrelor in stable CAD/COPD patients may be, at least in part, mediated by its capacity to reduce systemic inflammation and oxidative stress.

## 1. Introduction

Dual antiplatelet therapy (DAPT), consisting of the co-administration of aspirin and of a P2Y_12_ inhibitor, is the gold standard treatment for chronic stable coronary artery disease (CAD) patients following percutaneous coronary intervention (PCI). Among antiplatelet drugs, ticagrelor, due to its higher efficiency in inhibiting the P2Y_12_ receptor, is a more potent platelet inhibitor as compared with clopidogrel [1,2]. In acute coronary syndrome (ACS) patients, treatment with ticagrelor resulted in a reduced number of cardiovascular events and in improved peripheral arterial function compared to clopidogrel [3,4,5], suggesting that ticagrelor, beside its antiplatelet activity, modulates biological processes involved in cardiovascular protection. In patients with stable CAD and concomitant chronic obstructive pulmonary disease (COPD), we have shown that ticagrelor, compared with clopidogrel, is superior in improving biological markers of endothelial function [2,6]. These benefits are consistent with observations in patients with stable CAD and concomitant diabetes in which treatment with ticagrelor led to higher improvement of endothelial dysfunction in comparison to ticagrelor [7]. Lastly, unlike clopidogrel, ticagrelor retained some efficacy to prevent ischemic events over and above those of aspirin during chronic therapy [8].

To date, the molecular mechanisms underlying these pleiotropic effects of ticagrelor have not been elucidated, but there is evidence suggesting that they are related to inhibition of the equilibrative nucleoside transporter-1 (ENT1) leading to increased circulating levels of adenosine and cyclic adenosine monophosphate (cAMP) [9,10]. Adenosine, by binding adenosine receptors (ARs), regulates biological processes involved in CAD and COPD progression including oxidative stress and inflammatory responses [11], and functions of monocytes, neutrophils, CD4 T-helper cells, and CD8 cytotoxic T-cells, all implicated in CAD [12,13]. Specifically, in human monocytes, adenosine reduces inflammation through the inhibition of the expression of adhesion molecules, such as intercellular adhesion molecule 1 (*ICAM1*) [14]. Furthermore, in human monocyte-derived macrophages, cAMP limits inflammasome activation by promoting LR pyrin domain containing 3 (*NLRP3*) degradation [15,16], this latter found upregulated in cardiovascular disease [17]. Moreover, in several inflammatory diseases [18,19,20,21], A2aR signaling activation has been associated to the reduction of pro-inflammatory T-helper (Th) 17 cells and the increase of anti-inflammatory T-regulatory (Tregs) cells. Importantly, Th17/Tregs ratio has been found to be a determinant of both CAD and COPD progression [22,23,24,25]. Furthermore, adenosine, by downregulating the expression of Notch1 receptor, suppresses the activity of inflammatory CD8 T-cells [26], which are increased in COPD and stable CAD patients [27,28].

Dysregulation of the Notch signaling, a major regulator of cardiovascular homeostasis and of innate and adaptive immune response, has been observed in cardiovascular diseases [13,29]. Specifically, altered levels of Notch ligands JAGGED-1 (JAG1) and Delta-like 4 (DLL4) have been linked to the progression of atherosclerosis [30,31]. Of interest, altered expression levels of Notch1 and its target gene *HES1* have been associated to COPD [32,33]. The Notch pathway is regulated by cross-talks with a plethora of pathways [34,35] including the NAD^+^-dependent protein deacetylase, sirtuin1 (SIRT1), an oxidative stress sensor [36] and a repressor of inflammatory response [37], whose expression and activity is reduced in peripheral blood mononuclear cells (PBMCs) of stable CAD, ACS [38], and COPD [39] patients.

Many studies have investigated the possible molecular mechanisms underlying the pleiotropic activity of ticagrelor in ACS [40,41,42,43] and SCAD [7] providing evidence of a ticagrelor-mediated increase in circulating levels of adenosine and cAMP [10,44,45,46,47]. Additionally, several studies have also focused on the effects of ticagrelor on circulating inflammation markers [48,49,50,51]. The aim of our study was to expand the comparison between ticagrelor and clopidogrel to include the effects of treatment on markers of inflammation related to endothelial dysfunction (Appendix A) never investigated in this context, in order to provide more molecular details that could help to gain a better understanding of the pleiotropic effect of ticagrelor on the vascular system.

## 2. Results

### 2.1. Ticagrelor, But Not Clopidogrel, Increases SIRT1 and HES1 mRNA Levels

We compared the mRNA expression levels of *RORγt* (Th17 cells transcription factor), *FoxP3* (Tregs transcription factor), *NLRP3*, *ICAM1*, *SIRT1*, Notch ligands *JAG1* and *DLL4,* and of *HES1*, Notch target gene, in RNA isolated from peripheral blood cells obtained from stable CAD/COPD patients following PCI before (T0) and after 1-month (T30) treatment with ticagrelor or clopidogrel. At T0, mRNA levels of *RORγt, FoxP3*, *NLRP3*, *ICAM1,* and *JAG1* did not differ between the treatment groups (Figure 1A–E), whereas *HES1* mRNA levels were lower in ticagrelor, compared to clopidogrel (Figure 2B). In both ticagrelor- and clopidogrel-treated groups, we found no significant differences between T0 and T30 in the levels of *RORγt*, *FoxP3*, *NLRP3*, *ICAM1,* and *JAG1* mRNAs (Figure 1A–E). The results relative to *DLL4* mRNA are not shown because the expression level of this ligand was too low to provide a reliable assessment of the expression of this gene.

*SIRT1* and *HES1* mRNAs were significantly increased at T30 compared to T0 following ticagrelor but not clopidogrel treatment (Figure 2A,B). Before–after analyses showed that unchanged levels of *RORγt*, *FoxP3*, *NLRP3*, *ICAM1,* and *JAG1* mRNAs between T0 and T30 were not due to lack of response of these genes expression to each drug, but rather to a heterogeneous response characterized by similar number of patients showing no difference, increased or reduced levels of the specific mRNA in response to treatment (Appendix A). Instead, the response of *SIRT1* to ticagrelor showed prevalently no changes (10/20 patients) or increased (8/20 patients) mRNA levels and reduced levels only in 2/20 patients. On the contrary, in the clopidogrel group the changes in *SIRT1* mRNA were equally distributed between patients showing no changes (8/21), increased (6/21), or reduced levels (7/21) of mRNA (Figure 3A). Similarly, in the ticagrelor group *HES1* mRNA increased in 15/21 patients and was unchanged in 1/21 patients or decreased in 5/21 patients. In the clopidogrel-treated group the levels of *HES1* mRNA were increased in 8/21 patients, decreased in 9/21 patients, or unchanged in 4/21 patients (Figure 3B).

Severity of diseases, co-morbidities, and pharmacological treatment can determine differences in gene expression. Furthermore, drug interaction may modulate ticagrelor concentration and therefore effectiveness of the treatment [52]. To identify possible associations between disease severity, comorbidity or drug treatment, we performed correlation analyses between these variables and changes in *SIRT1* or *HES1* after a 1-month treatment. Interestingly, we found that being a man is associated with an increase in *SIRT1* expression after treatment (R = 0.357; *p* = 0.022). We also found an inverse association between the variations of *SIRT1* and treatment with inhaled corticosteroids (R = −0339; *p* = 0.003) or long-acting beta2 agonist (R = −0.311; *p* = 0.048) (Appendix A). These data show that men respond better than women to ticagrelor with regard to *SIRT1* expression and suggest that drugs for the management of COPD may affect the effectiveness of treatment with ticagrelor. On the contrary, we found no association between other co-morbidities, COPD severity, and treatment response. Due to the high homogeneity of patients with stable CAD and the relative small study population, it was not possible to stratify patients according to CAD severity.

### 2.2. Correlation Analyses Between SIRT1 and HES1 mRNA Levels and Markers of Endothelial Dysfunction in Treated Patients

Endothelial dysfunction (ED) comprises alterations of the endothelial cell (EC) physiology (including increased expression of pro-inflammatory cell adhesion molecules (CAMs), such as intercellular adhesion molecule-1 (ICAM1) and vascular cell adhesion molecule-1 (VCAM1), impaired nitric oxide (NO) production and signaling, EC apoptosis, and increased vascular permeability) that can be reflected in changes in the serum of patients. We have developed an ex-vivo assay to determine the levels of apoptosis in the endothelium of patients. The assay involves the determination of the number of annexin V/propidium iodide positive apoptotic human umbilical vein endothelial cells (HUVECs) following 48 h of cultivation in the presence of medium containing 20% of patients’ serum. We have shown that the levels of apoptosis in HUVECs treated with serum from stable and acute CAD patients reflect the gravity of the disease [53,54]. Furthermore, we have reported that this assay can reliably detect a reduction in apoptosis in HUVECs treated with serum from stable CAD/COPD patients following treatment with ticagrelor [6] or from subjects with hypercholesterolemia and low-moderate cardiovascular risk, following eight weeks consumption of a red yeast rice, polymethoxyflavones, and antioxidants containing nutraceutical supplement [55]. In another study by our group, this approach was used, in parallel with measurements of changes in brachial wall shear stress and flow mediated dilation, to assess ED in patients that underwent surgical or transcatheter aortic valve replacement [56]. Of relevance, in stable CAD/COPD patients, following ticagrelor treatment we also found decreased serum levels of epidermal growth factor (EGF), increased NO generation in HUVECs treated with patient’s serum, measured by diaminofluorescein (DAF) assay, and attenuated reactive oxygen species (ROS) production in PBMC isolated from patients, assessed by flow cytometric analysis [2,6]. In order to determine whether the observed changes of *SIRT1* and *HES1* mRNA could be related to these markers of endothelial function, and whether the different effect of the drugs on *SIRT1* and *HES1* mRNA could be explained by differences in antiplatelet activity, we conducted association analyses between changes in *SIRT1* and *HES1* mRNA and changes in the levels of EGF, ROS, apoptosis, NO and platelets reactive units (PRU) after 1-month treatment (changes were defined as the ratio between values at T30 over values at T0).

Following treatment with ticagrelor or clopidogrel, we found correlations between changes in *HES1* and *SIRT1* mRNA levels and changes in the levels of EGF (*HES1*, R = −0.469, *p* < 0.01; *SIRT1*, R = −0.440, *p* < 0.01) (Figure 4A). In addition, changes in *HES1* mRNA levels also correlated with changes in *SIRT1* mRNA levels (R = 0.466, *p* < 0.01) (Figure 4B). On the contrary, we found no correlation among changes in *HES1* or *SIRT1* and ROS (*HES1*, R = −0.150, *p* = 0.36; *SIRT1*, R = −0.78, *p* = 0.65), apoptosis (*HES1*, R = 0.176, *p* = 0.27; *SIRT1*, R = 0.077, *p* = 0.63) and NO (*HES1*, R = 0.002, *p* = 0.99; *SIRT1*, R = −0.05, *p* = 0.76). Similarly, no correlation was found between changes in *HES1* or *SIRT1* mRNA and changes in the levels of PRU (*HES1*, R = −0.163, *p* = 0.31; *SIRT1*, R = −0.207, *p* = 0.20).

## 3. Discussion

Although concerns remain as to dosage equivalence, the PLATO (PLATelet inhibition and clinical Outcome) trial results showed that ticagrelor is superior to clopidogrel in preventing ischemic events in patients with acute coronary syndrome (ACS) [5]. To test the effect of ticagrelor over clopidogrel in patients with CAD, we selected a population of patients with stable CAD and chronic inflammation supported by the presence of a COPD diagnosis. COPD is a well-known chronic condition associated with inflammation and ED [57,58]. A similar approach has already been used in the study of Mangiacapra et al. who enrolled patients with stable CAD and type 2 diabetes to study the effect of ticagrelor and clopidogrel on endothelial function. In this case, diabetic patients were selected because they are a well-established subgroup of patients with ED, independently by cardiologic clinical presentation [7]. The choice to investigate the effects of ticagrelor on ED is in apparent contrast with a study showing that the anti-aggregatory effects of ticagrelor are less pronounced during acute inflammation [59], but is supported by data showing pleiotropic effects of ticagrelor beyond its potent antiplatelet effects [60,61]. Consistently, we found reduced levels of surrogate markers of endothelial dysfunction in patients with stable CAD and concomitant COPD following PCI and 1-month treatment with ticagrelor, but not with clopidogrel [6]. We now report that, in the same patients, treatment with ticagrelor, but not with clopidogrel, results in increased of levels of *SIRT1* and *HES1* mRNAs in peripheral blood cells.

Ticagrelor, unlike clopidogrel, has antiplatelet- and P2Y_12_-independent activities that seem to involve increased levels of adenosine/cAMP in plasma able to reduce both systemic inflammation and oxidative stress [9,10], although a single study has shown that ticagrelor is not able to increase adenosine circulating levels [62]. In order to gain more insights on the molecular pathways modulated by ticagrelor, we determined, in RNA isolated from peripheral blood cells of stable CAD patients following PCI, the effect of 1-month treatment with ticagrelor or clopidogrel on mRNA expression levels of inflammation- and oxidative stress-related genes, such as *ICAM1*, *NLRP3*, *SIRT1*, *JAG1*, *DLL4*, *HES1*, *RORγt* (Th17 marker), and *FoxP3* (Tregs marker). Among the mRNAs analyzed, we found an increase of *SIRT1* mRNA levels after treatment with ticagrelor but not clopidogrel. Specifically, we found that, although the number of patients with increased levels of *SIRT1* were comparable in both treatment groups (6/21 in clopidogrel and 8/20 in ticagrelor), in the ticagrelor-treated group, only 2/20 patients showed a significant reduction of *SIRT1* levels, which instead was detected in 7/21 patients treated with clopidogrel (Figure 3A). Our results, suggesting that ticagrelor may counteract *SIRT1* reduction in RNA from peripheral blood of stable CAD/COPD patients, are consistent with previous reports of reduced activity of SIRT1 in serum of COPD patients [39] and inhibition of *SIRT1* gene in peripheral monocytes in ACS and stable CAD patients [38]. SIRT1, one of the class III NAD^+^-deacetylases sirtuin family [63], has gained great attention for its impact on the regulation of aging-related processes [36]. *SIRT1* blunts inflammatory response and oxidative stress mainly through the deacetylation of target proteins, such as Forkhead box O3 A (Foxo3a) [64], activator protein-1 (AP-1) [65], nuclear factor kappa B (NF-κB) [66], Notch1 [67,68], and NLRP3 [69]. Furthermore, SIRT1 interferes with Toll-like receptor (TLR) 2- mediated monocyte adhesion to the vascular endothelium, an early step in the pathogenesis of CAD [70]. Of interest, *SIRT1* genetic overexpression significantly decreased oxidative stress in cigarette smoke-exposed rodent lungs [71].

We also found increased expression of *HES1* in more than a half of ticagrelor-treated patients (15/21), compared to the clopidogrel group, in which the number of patients with increased or decreased *HES1* mRNA levels were similar (8/21 and 9/21, up- and down-regulated, respectively) (Figure 3B). *HES1*, a transcription factor belonging to the Hairy and Enhancer of Split (HES) family, is required for differentiation of T and B cells [72] and plays a major role in the reduction of inflammation and neutrophil-mediated responses by controlling production of macrophage-derived chemokines [73,74]. In macrophages, *HES1* functions as a feedback inhibitor of production of pro-inflammatory cytokines, such as IL (Interleukin)-6 and IL-12 [75,76]. *HES1* is a target gene and essential transducer of the Notch pathway, implicated in the regulation of cell fate decisions, such as differentiation, proliferation and survival [77]. The Notch pathway is triggered by the interaction between the Notch receptors (Notch 1–4) and ligands (DLL1, 3, 4 and JAGGED-1, -2), on adjacent cells, leading to the release of the active form of Notch which translocates into the nucleus, where it promotes the transcription of its target genes [77]. Notch regulates heart and vascular functions during development and is involved in the repair of the damaged and/or stressed myocardium [29,78,79]. Dysregulation of the Notch pathway has been linked to atherosclerosis [80,81] and to COPD [82]. In COPD patients, reduced levels of *NOTCH1* and *HES1* have been shown in the endothelium [33] and in the airways epithelium [32].

Our data, analyzed in the context of the existing literature, suggest that increased levels of *SIRT1* and *HES1* in RNA of peripheral blood cells of stable CAD/COPD patients may be part of the protective action of ticagrelor. Of interest, we found a positive correlation between changes of *SIRT1* and *HES1* mRNA levels after the treatment with ticagrelor or clopidogrel (Figure 4B). Little is known about *SIRT1* and *HES1* interaction. *SIRT1* physically interacts with *HES1* leading to stimulation of its role as negative regulator of transcription [83], whereas in neural stem cells *SIRT1* through inhibition of *HES1* expression mediates response of the cell to glucose [84] and induces their differentiation [85]. The biological meaning of this association in the context of ticagrelor/clopidogrel treated stable CAD/COPD patients deserves further studies.

Correlation analysis showed that *SIRT1* and *HES1* changes were not associated to changes in platelet activity, indicating that the ticagrelor effect on these parameters is not directly related to the inhibition of P2Y_12_. On the contrary, changes in both mRNA *SIRT1* and *HES1* levels negatively correlated with changes in EGF levels (Figure 4A). High levels of EGF and its receptor EGFR have been detected in lung tissue specimens of COPD patients [86] and high airway immunoreactivity has been associated to mucin hypersecretion in COPD [87]. EGF is involved in endothelial dysfunction, neointimal hyperplasia, cardiac hypertrophy and remodeling [88] and, as shown by us, reduction in circulating levels of EGF could be involved in the ticagrelor-mediated improvement of endothelial function in stable CAD/COPD patients [2]. The cross-talk between EGF and Notch signaling is well characterized: EGF and Notch pathways can cooperate in either synergistic or antagonistic fashion in malignancies [89] and the EGF, via EGFR, acts as a negative regulator of *HES1* expression in keratinocytes [90]. On the contrary, little is known on the cross-talk between EGF/*SIRT1*. In cancer, *SIRT1* is upregulated during EGF-mediated epithelial-to-mesenchymal transition [91], while in vascular smooth muscle cells, resveratrol, a *SIRT1* stimulator, interferes with EGF-induced ROS production [92]. The negative correlation found in this study may be suggestive of a cross-talk among EGF, *SIRT1,* and/or *HES1* in peripheral blood of stable CAD/COPD patients, according to which ticagrelor-mediated EGF-reduction may lead to the induction of *SIRT1*/*HES1* axis.

A limitation of this study is represented by the use of RNA isolated from whole peripheral blood, which contains different cell types (lymphocytes, monocytes, and granulocytes). Therefore, no conclusions can be drawn with respect to the molecular mechanism involved in *SIRT1* and *HES1* upregulation, which could be due to the effect of the drug either on the expression of these genes or on the size of specific populations of cells expressing these genes. The use of whole blood could have also reduced the sensitivity of our assay, thus allowing the detection, at follow up, of significant changes in the expression of *SIRT1* and *HES1* but of none of the other genes analyzed (*RORγt*, *FoxP3*, *NLRP3*, *ICAM1*, *JAG1,* and *DLL4*).

Another limitation is the lack of information on the deacetylase activity of SIRT1 in our samples that, as shown by Conti and collaborators, would have provided more specific information, compared to the mRNA, about the role of *SIRT1* [39] in mediating the pleiotropic effects of ticagrelor in stable CAD/COPD patients. This limitation could explain the lack of association between changes in *SIRT1* mRNA and neither the other inflammation markers investigated (*NLRP3*, *ICAM1*, *RORγt,* and *FoxP3*), known to be regulated by *SIRT1* [64,66,67,69] nor the surrogate markers of endothelial function (apoptosis, ROS and NO levels), previously found by us to be reduced by ticagrelor treatment [6].

A third limitation of the study is represented by the relatively small number of patients and by the impossibility to conduct statistical power analysis due to lack of published data on the effects of ticagrelor/clopidogrel on the expression of the studied genes. Therefore, this should be considered a pilot study, providing information for future studies aimed to validate our findings, which could be relevant not only for dissecting the mechanism of action of ticagrelor but also for providing novel blood markers (*SIRT1* and *HES1* mRNA) able to predict whether the single patient will benefit of effects of ticagrelor beyond its antiplatelet activity.

## 4. Material and Methods

### 4.1. Study Design and Population/Randomization and Interventions

This is a sub-study of the clinical trial “The comparisoN between ticAgrelor and clopidogrel effect on endoTHelial platelet ANd iNflammation parameters in patiEnts with stable coronary artery disease and chronic obstructiVE pulmonaRy disease undergoing percutaneous coronary intervention (NATHAN-NEVER)” registered at www.clinicaltrials.gov (NCT02519608, September 2015). The protocol was approved by “Comitato Etico Unico della Provincia di Ferrara”. Written informed consent was given by all subjects in accordance with the Declaration of Helsinki. This study was an investigator-initiated, prospective, single-center, randomized, open-label phase II trial involving 46 consecutive patients with stable coronary artery disease requiring coronary artery angiography (CAA) and PCI, and concomitant COPD. The diagnosis of COPD was based on spirometry data according to international guidelines and in particular by the presence of not fully reversible airflow limitation defined as a ratio between forced expiratory volume at 1 s (FEV1) and forced vital capacity (FVC) ratio < 0.7 after the administration of a bronchodilator [93]. An independent blinded reviewer analyzed spirometry and documentation available to confirm COPD diagnosis and to adjudicate severity of airflow limitation based on post-bronchodilator FEV1 (from mild to very severe). Patients were randomly assigned to receive clopidogrel (*n* = 23) or ticagrelor (*n* = 23) on top of standard therapy with aspirin. The primary endpoint of the study was the 1-month rate of HUVECs apoptosis and the rate of apoptosis after 1 month was significantly lower in patients treated with ticagrelor (7.4 ± 1.3% vs. 9.3 ± 1.5%, *p* < 0.001), satisfying the pre-specified primary endpoint [6]. The study design, the characteristics of the study participants, including the severity of both diseases, comorbidities, and drug treatment, the outcomes of this trial are described in [6].

### 4.2. Blood Samples

A 21-gauge needle was used to collect blood samples from an antecubital vein of patients before PCI and drug administration, and at the 1-month visit. Details of the sampling procedures are reported in [6]. After the withdrawal, blood samples were stored frozen until RNA extraction.

### 4.3. RNA Isolation and cDNA Synthesis

RNA was isolated from blood samples using the QIAamp RNA Blood Mini Kit (Qiagen, Carlsbad, CA, USA). One volume of blood was mixed with 5 volumes of buffer EL (erythrocyte lysis) and incubated for 10 to 15 min on ice, vortexing briefly 2 times during incubation to selectively lyse red blood cells. White cells were then collected by centrifugation at 400× *g* for 10 min at 4 °C and supernatant was removed. White cells were lysed using highly denaturing conditions to inactivate the RNases. After homogenization using the QIAshredder spin column, the sample was applied to the QIAamp spin column. Total RNA binds to the QIAamp membrane and contaminants are washed away, leaving pure RNA to be eluted in 30–100 µL RNase-free water. RNA concentration and purity were determined by NanoDrop 2000 spectrophotometer (Thermo Fisher Scientific, Waltham, MA, USA). RNA was treated with the RNase-Free DNase Set (Qiagen, Carlsbad, CA, USA), in order to eliminate DNA contamination. The 100 ng of total RNA were reverse transcribed to cDNA using the SuperScript™ III First-Strand Synthesis SuperMix (Life Technologies, Carlsbad, CA, USA).

### 4.4. Droplet Digital PCR Reaction

For droplet digital (dd) PCR, 2 µL of cDNA (1 ng/µL) were used in each reaction. 20 µL of a solution containing cDNA, primers and QX200™ ddPCR™ EvaGreen Supermix (Bio-Rad, Hercules, CA, USA) were used for droplets preparation using a Bio-Rad QX200 droplet generator (Bio-Rad, Hercules, CA, USA). Emulsified samples were transferred to a 96-well plate, sealed with the PX1 PCR plate sealer (Bio-Rad, Hercules, CA, USA) and amplified using the SimpliAmp thermal cycler, (Applied Biosystems, Waltham, CA, USA). The parameters for ddPCR were: 95 °C enzyme activation step for 5 min followed by 40 cycles of a two-step cycling protocol (95 °C for 30 s and 60 °C for 1 min). The sequences of primers used are shown in Appendix A. At the end of thermal cycling, a QX200 droplet reader (Bio-Rad, Hercules, CA, USA) was used to quantify the number of generated droplets from each reaction mix. Expression data was extracted using QuantaSoft software (Bio-Rad, Hercules, CA, USA) and the absolute quantity of cDNA per sample (copies/µL) was normalized to the average number of copies of *GUSB* (β-glucuronidase) mRNA in each sample.

### 4.5. Statistical Analysis

Normal distribution of the variables was verified with the D’Agostino–Pearson normality test and with the Shapiro–Wilk test (alpha = 0.05). Variables were presented as scatter plot with median. Outliers were identified and removed. Normally distributed variables were checked for homoscedasticity by Levene test. Follow up toward baseline groups were compared employing repeated measures ANOVA to take into account for samples dependency. Student t test for independent measures was employed to test statistical significance in the mean differences of gene expression levels at baselines between treatments. *p*-values ≤ 0.05 were considered statistically significant. Correlations between continuous variables were tested by Spearman’s correlation coefficient correlation. Correlation of baseline categorical variables and gene expression was tested by point-biserial correlation. Statistical analysis was performed with GraphPad Prism version 7.0 (GraphPad software Inc., San Diego, CA, USA) and R (R Foundation for Statistical Computing, Vienna, Austria, 2019).

## 5. Conclusions

In conclusion, we showed that the effect of ticagrelor related to the mitigation of endothelial dysfunction in stable CAD/COPD patients may be mediated, at least in part, by its capacity to increase *SIRT1* and *HES1* mRNAs levels (Appendix A). These findings are consistent with studies suggesting that the pleiotropic effects of ticagrelor on the vascular system may be ascribed to a reduction of systemic inflammation and oxidative stress.

## Figures and Tables

**Figure 1 ijms-21-01576-f001:**
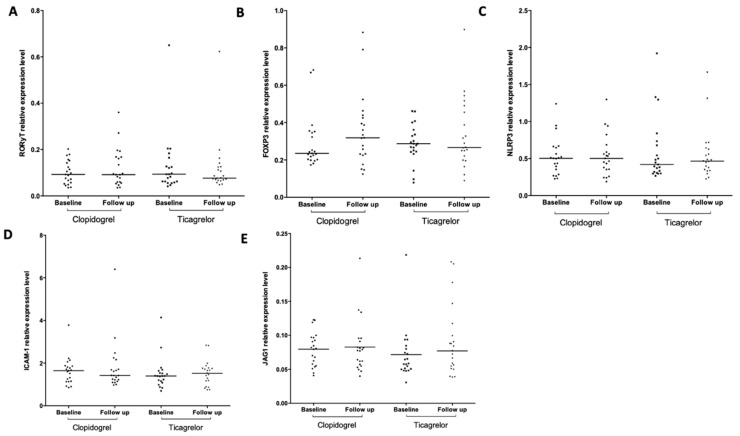
Droplet digital (dd) PCR based analysis of the expression of inflammation- and oxidative stress-related genes in peripheral blood cells of stable coronary artery disease (CAD)/concomitant chronic obstructive pulmonary disease (COPD) patients following 1-month treatment with ticagrelor and clopidogrel. Scatter plots, with medians, of the expression levels of *RORγt* (**A**), *FoxP3* (**B**), *NLRP3* (**C**), *ICAM1* (**D**), and *JAG1* (**E**) are shown. The absolute quantity of cDNA (copies/µL) was normalized to the average number of copies of *GUSB*. Follow up vs. baseline gene expression values, ANOVA test. Comparison of gene expression levels at baseline, student t test.

**Figure 2 ijms-21-01576-f002:**
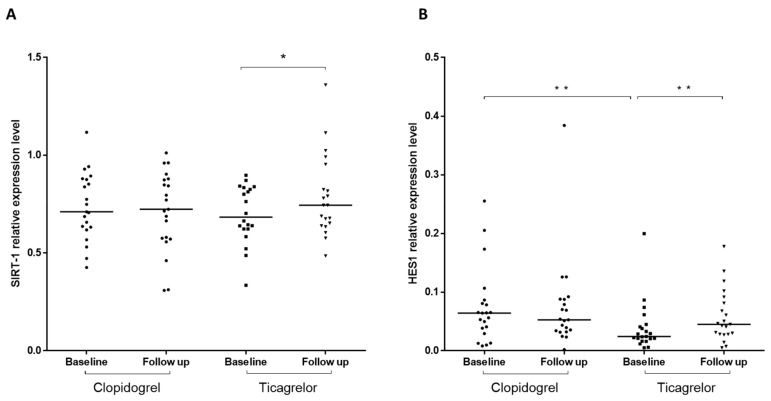
Droplet digital (dd) PCR based analysis of the levels of *SIRT1* and *HES1* mRNA in peripheral blood cells of stable CAD/COPD patients following 1-month treatment with ticagrelor or clopidogrel. Scatter plots, with medians, of the expression levels of *SIRT1* (**A**) and *HES1* (**B**) in peripheral blood cells of stable CAD/COPD patients following 1-month treatment with ticagrelor or clopidogrel. The absolute quantity of cDNA (copies/µL) was normalized to the average number of copies of *GUSB*. Follow up vs. baseline gene expression values, ANOVA test, * *p* < 0.05 and ** *p* < 0.01. Comparison of gene expression levels at baseline, student t test, ** *p* < 0.01.

**Figure 3 ijms-21-01576-f003:**
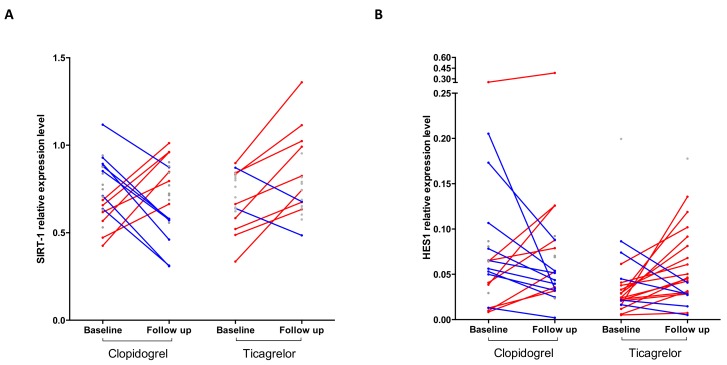
Before-after analysis of *SIRT1* and *HES1*. Before-after plot of *SIRT1* (**A**) and *HES1* (**B**) gene expression in peripheral blood cells from patients before and after one month of treatment with clopidogrel or ticagrelor. For clarity, only changes in gene expression higher than 20% of the values at baseline are connected and color-coded (red for fold changes >1.2 and blue for <0.8).

**Figure 4 ijms-21-01576-f004:**
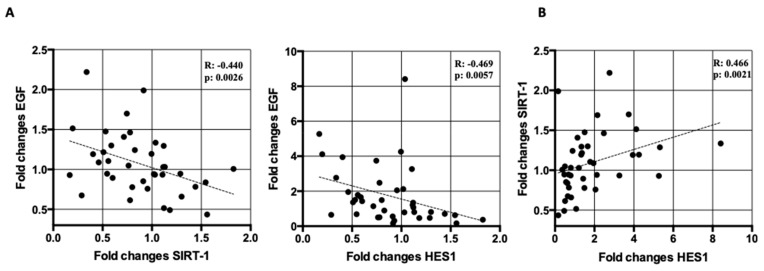
Correlation analyses. (**A**) Correlation between changes in *SIRT1* and *HES1* mRNA expression levels and changes in serum levels of Epidermal Growth Factor (EGF). Correlations were assessed by Spearman’s correlation test (*HES1*, R = −0.469, *p* < 0.01; *SIRT1*, R = −0.440, *p* < 0.01). (**B**) Correlation between changes in *HES1* and *SIRT1* mRNA expression levels. Correlations were assessed by Spearman’s correlation test (R = 0.466, *p* < 0.01). *SIRT1* and *HES1* changes were defined as ratio between mRNA levels after (T30) and before (T0) treatment with ticagrelor or clopidogrel. EGF changes were expressed as ratio between serum EGF concentration at T30 over serum EGF concentration at T0.

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
