# Peer review of "Ticagrelor Increases SIRT1 and HES1 mRNA Levels in Peripheral Blood Cells from Patients with Stable Coronary Artery Disease and Chronic Obstructive Pulmonary Disease"

_ijms, 2020, doi:10.3390/ijms21051576_

Round 1
Reviewer 1 Report
I want to express my gratitude for the opportunity to re-review the manuscript entitled "Ticagrelor increases SIRT1 and HES1 mRNA levels in peripheral blood cells from patients with stable coronary artery disease and chronic obstructive pulmonary disease". The Authors took into account my previous comments regarding the construction of the introduction and discussion, as well as the presentation of methods and results. Also, the supplementary data do not raise my objections. To sum up, in my opinion, the manuscript meets the criteria necessary for publication in IJMS.
Author Response
We thank the Reviewer for the kind comments and constructive criticism. Our "point-to-point" response to the Reviewer comments is as following:
Response to Reviewer 1 Comments
I want to express my gratitude for the opportunity to re-review the manuscript entitled "Ticagrelor increases SIRT1 and HES1 mRNA levels in peripheral blood cells from patients with stable coronary artery disease and chronic obstructive pulmonary disease". The Authors took into account my previous comments regarding the construction of the introduction and discussion, as well as the presentation of methods and results. Also, the supplementary data do not raise my objections. To sum up, in my opinion, the manuscript meets the criteria necessary for publication in IJMS.
Response: We thank the Reviewer for the nice comment about our work and we are grateful for his/her
constructive criticism that helped us to improve the manuscript.
Reviewer 2 Report
The various revisions made in this manuscript enhance its scientific merit, and in general I am happy with them.
A few changes are necessary:-
(1)line 64: it is not true that the PEGASUS trial demonstrated superiority of ticagrelor over clopidogrel during chronic therapy. This sentence should be deleted, or changed to say that, unlike clopidogrel, ticagrelor retains some efficacy to prevent ischaemic events over and above those of aspirin during chronic therapy.
(2)line 345: Reference 57 seems inappropriate here: surely the PLATO trial is the correct reference? Even so, there are prsisting arguments about dosage equivalence. Perhaps better would be:
"Although concerns remain as to dosage equivalence, the PLATO trial results suggested that..."
(3)Line 285: Change to..."impired niric oxide(NO) production and signalling.."
(4) Discussion: The concept that ticagrelor effects may be more pronounced in the presence of inflammatory activation, as provided by background respiratory disease, is interesting. Perhaps it should be added that this contrasts with the anti-aggregatory effects of ticagrelor, which are LESS pronounced during acute inflammation as exmplified by acute coronary syndromes (Imam et al, Thrombosis Research,2019).
Author Response
We thank the Reviewer for the kind comments and constructive criticism. Our "point-to-point" responses to the Reviewer comments are as following:
Response to Reviewer 2 Comments
The various revisions made in this manuscript enhance its scientific merit, and in general I am happy with them.
We thank the Reviewer for the nice comment about our work and we are grateful for his/her constructive criticism
A few changes are necessary:-
Point 1: line 64: it is not true that the PEGASUS trial demonstrated superiority of ticagrelor over clopidogrel during chronic therapy. This sentence should be deleted, or changed to say that, unlike clopidogrel, ticagrelor retains some efficacy to prevent ischaemic events over and above those of aspirin during chronic therapy.
Response 1: In accordance to the Reviewer’s comment we modified the text as follow: “Lastly, unlike clopidogrel, ticagrelor retained some efficacy to prevent ischemic events over and above those of aspirin during chronic therapy”
Point 2: line 345: Reference 57 seems inappropriate here: surely the PLATO trial is the correct reference? Even so, there are prsisting arguments about dosage equivalence. Perhaps better would be:
"Although concerns remain as to dosage equivalence, the PLATO trial results suggested that..."
Response 2: We modified the text as suggested in the Reviewer’s comment. We replaced reference [57] (Valgimigli, M. et al.) with Wallentin, L. et al. (N Engl J Med 2009, 361, 1045-1057), already cited into the manuscript.
Point 3: Line 285: Change to..."impired niric oxide(NO) production and signalling.."
Response 3: We added “signaling” in the text as indicated by the Reviewer.
Point 4: Discussion: The concept that ticagrelor effects may be more pronounced in the presence of inflammatory activation, as provided by background respiratory disease, is interesting. Perhaps it should be added that this contrasts with the anti-aggregatory effects of ticagrelor, which are LESS pronounced during acute inflammation as exmplified by acute coronary syndromes (Imam et al, Thrombosis Research,2019).
Response 4: Following the Reviewer’s comment we modified the text as follow (line 206): “The choice to investigate the effects of ticagrelor on ED is in apparent contrast with a study showing that the anti-aggregatory effects of ticagrelor are less pronounced during acute inflammation (Imam, Nguyen et al. 2019) but is supported by data showing pleiotropic effects of ticagrelor beyond its potent antiplatelet effects (Jeong, Hong et al. 2017, Zheng, Li et al. 2019).”
This manuscript is a resubmission of an earlier submission. The following is a list of the peer review reports and author responses from that submission.
Round 1
Reviewer 1 Report
The central idea behind this manuscript is that ticagrelor, rather than clopidogrel, exerts anti-inflammatory effects in patients who have stable coronary disease (and presumably angina pectoris) PLUS chronic obstructive pulmonary disease (COPD).
My first serious concern about the study is that there is no obvious clinical relevance for either the study or the specification of concomitant COPD. No particular clinical benefit has ever been shown for ticagrelor in the context of COPD, and if the authors are looking for evidence of an important anti-inflammatory effect of ticagrelor to benefit coronary outcomes, surely it would be better to study coronary disease (definitely in part an inflammatory process!) by itself. Second, there are major clinical concerns about the use of ticagrelor in chronic stable ischaemia: it has never been compared with clopidogrel in this context (and really this comparison was performed only in the PLATO study, where there was a concern about non-equivalent dosing of ticagrelor versus clopidogrel, so even in the contet of acute coronary syndromes, the comparison remains a little questionable, and there is NOTHING relevant in chronic disease context. Indeed, the use of long-term ticagrelor in the PEGASUS study, involving patients with chronic coronary disease, was noteworthy for a very nasty long-term bleeding rate. All this information is not really made clear in the manuscript.
My next category of concern is pharmacological: while it is true that ticagrelor is a drug with pleiotropic actions, and notably its interaction with adenosine receptors, it would have been relatively easy in the current, essentially ex vivo investigation to determine whether adenosine receptor interactions might have mediated any of the reported effects. To say that "ticagrelor is a more potent P2Y12 antagonist than clopidogrel is seriously misleading: it is more potent on a weight-for-weight basis, but it is a comparatively easy matter to achieve equivalent dosing, and this is not mentioned anywhere.
My next category of serious concern relates to the source of these data, the clinical study described in reference 6. I think that this manuscript representing the third sub-analysis of this rather small study (which randomized only 23 patients to each treatment arm and did not actually measure vascular endothelial function in any sort of conventional way). I am not sure that such partitioning of data is to b encouraged, nor is it reasonable scientifically to divide a single-purpose study thus. Of course it is fine to measure "biological measures of endothelial function", but why not measure PHYSIOLOGICAL ones, and why not use a blinded structure when doing so?? And if we were to need core biochemical data, why not measure plasma cAMP concentrations (key to the effects of both ticagrelor and clopidogrel) and also plasma ADMA concentrations as major modulators of NOS effect?
Next, the experimental methodology is quite vague. For example, RNA isolation was from "peripheral blood cells". Which cells, please? Why are the details of none of the actual assays provided? As regards the statistics, could we please see the results of ANOVAs (the key statistics used) as well as post hoc tests?
I cannot see any merit in the statement (Discussion) that "SIRT1 and HES1 changes were not associated to platelet activity".. As I understand it, only ON_TREATMENT platelet aggregability was measured. Such aggregability would reflect an unknown combination of drug effect and initial platelet reactivity, not a logical comparator: the right comparaor would be drug-induced change in platelet reactivity.
The last paragraph of the Discussion, related to the "protective effect...", seems scientifically unwarranted by the totality of data.
Finally, I would like to remind the authors that SCAD is used as an acronym conventionally for "spontaneous coronary artery dissection", so perhaps they should re-think this.
Reviewer 2 Report
Reviewer's report
Manuscript title:
Ticagrelor increases SIRT1 and HES1 mRNA levels in peripheral blood cells from patients with stable coronary artery disease and chronic obstructive pulmonary disease
Authors:
Giorgio Aquila, Francesco Vieceli Dalla Sega, Luisa Marracino, Rita Pavasini, Laura Sofia Cardelli, Anna Piredda, Alessandra Scoccia, Valeria Martino, Francesca Fortini, Ilaria Bononi, Fernanda Martini, Antonio Pannuti, Paola Rizzo and Gianluca Campo
Inhibiting the P2Y12 platelet receptor is a primary therapeutic strategy in a stable coronary artery disease (SCAD) after percutaneous coronary intervention. Ticagrelor, in contrast to other P2Y12-receptor inhibitors, ensures rapid, potent, and consistent inhibition of platelet aggregation but also exerts other actions, beneficial in the context of atherosclerosis prevention and endothelium function. However, these effects cannot be solely explained by the P2Y12-receptor inhibition and result from other mechanisms of ticagrelor action that remain mostly unknown. Elucidation of these mechanisms may allow for the wider therapeutic use of ticagrelor. Therefore I find the paper interesting; however, I have some remarks regarding the data presentation and article structure.
Major Remarks
1) Introduction:
- the reasons for the examination of ticagrelor superiority over clopidogrel in the specific group of patients with co-existing SCAD and chronic obstructive pulmonary disease (COPD) should be better addressed.
- since the Authors examine the expression of several genes involved in the pathogenesis of atherosclerosis, endothelium dysfunction and oxidative stress a Figure presenting the interactions between the examined genes would be helpful to understand the objectives of the study
2) Methods:
- this part misses the clinical characteristics of the study participants (that could be placed in the result section as well), including the severity of both diseases, co-morbidities, and pharmacological treatment. This data is crucial for the proper interpretation of the results, especially when substantial inter-individual differences in the level of gene expression are observed. For instance, a drug interaction is a type of the confounder that may modulate ticagrelor concentration and therefore, effectiveness (as it was observed, e.g., in case of diltiazem, Teng R & Butler K. J Drug Assess. 2013;2:30–9.)
- since the number of study participants is limited, power calculation should be performed to assess the reliability of the results
3) Results:
- given the relatively small number of experiments, that the number of figures is relatively high. Moreover, the same data is presented in 3 ways: on graphs (Figures 3 & 4), in the table (part of Figure 3), and described in the text. To avoid repetitions, I would suggest to include Figure 3 in the supplementary data.
- it is not clear what parameters (apart from EGF) were assessed as “markers of endothelial dysfunction” (Part 3.2 in the results section) and what methodology was applied for their measurements
4) Discussion
- the limited number of study participants should be mentioned as an important limitation of the study. If the clinical characteristics of the patients are not available, this aspect should be added to the list of study limitations as well.
Minor Revision
The structure of the manuscript does not meet IJMS guidelines for research papers (namely: Introduction, Results, Discussion, Materials, and Methods); the same concerns the References. For details see: https://www.mdpi.com/journal/ijms/instructions
names of the genes should be written in italics